# Virulence and Genetic Types of *Blumeria graminis* f. sp. *hordei* in Tibet and Surrounding Areas

**DOI:** 10.3390/jof9030363

**Published:** 2023-03-16

**Authors:** Yunjing Wang, Qucuo Zhuoma, Zhi Xu, Yunliang Peng, Mu Wang

**Affiliations:** 1Research Institude of Tibet Plateau Ecology, Plant Sciences College, Tibet Agricultural and Animal Husbandry University, Nyingchi 860000, China; choc1336@163.com (Y.W.); zm563065652@163.com (Q.Z.); 2Institute of Plant Protection, Sichuan Academy of Agricultural Sciences/MOA Key Laboratory for Integrated Management of Pest on Crops in Southwest China, Chengdu 610000, China; xuzhi0125@163.com

**Keywords:** barley, powdery mildew, differential lines, SNP, population expansion

## Abstract

Barley (*Hordeum vulgare* L.) is the most important cereal crop in the Qinghai-Tibet Plateau, and the yield has been seriously threatened by *Blumeria graminis* f. sp. *hordei* (*Bgh*) in recent years. To understand the virulence and genetic traits of different *Bgh* populations, 229 isolates of *Bgh* were collected from Tibet, Sichuan, Gansu and Yunnan provinces of China during 2020 and 2021, and their pathogenicity to 21 barley lines of different genotypes was assessed. A total of 132 virulent types were identified. The *Bgh* isolates from Yunnan showed the highest diversity in terms of virulence complexity (*Rci*) and genetic diversity (*KWm*), followed by those from Sichuan, Gansu, and Tibet, in that order. Single nucleotide polymorphism (SNP) in genes coding for alternative oxidase (AOX), protein kinase A (PKA), and protein phosphatase type 2A (PPA) were detected at seven polymorphic sites. Nine haplotypes (H1–H9) with an average haplotype diversity (*Hd*) and nucleotide diversity π of 0.564 and 0.00034, respectively, were observed. Of these, haplotypes H1 and H4 accounted for 88.8% of the isolates, and H4 was predominant in Tibet. Genetic diversity analysis using the STRUCTURE (K = 2) and AMOVE indicated that the inter-group variation accounted for 54.68%, and inter- and intra-population genotypic heterogeneity accounted for 23.90% and 21.42%, respectively. The results revealed the recent expansion of the *Bgh* population in Tibet, accompanied by an increase in virulence and a loss of genetic diversity.

## 1. Introduction

Barley (*Hordeum vulgare* L.) is the fourth largest cereal crop in the world [1,2]. In China, the area for barley cultivation averaged 8 Mha during 1914 to 1918, dropped to 0.85 Mha in 2004, and further declined to 0.50 Mha in 2007 [3]. In 1997~2000, the barley production was mainly concentrated in the middle and lower reaches of the Yangtze River and North China. Then, the main barley production areas translocated to Southwest and Northwest China, and Jiangsu, Yunnan, Sichuan, Gansu and Inner Mongolia became the top five provinces for barley production in China [4]. Accounting for nearly 40% of the barley cultivation area in China, Yunnan, Sichuan and Tibet in the southwest are neighbored by Qinghai and Gansu in the northwest, geographically segregated from other regions of scaled barley production in China [5]. In the Tibet-Qinghai plateau and the surrounding Sichuan, Gansu, and Yunnan, the production areas covered approximately 0.03 Mha [6,7]. Unlike other types of barley mainly used for fodder and in the brewery industry, the hull-less barley (*H. vulgare* L. var. *nudum*), known as Qingke, is more suitable to the harsh environment in the cool, high altitude regions of the Qinghai-Tibet Plateau, and is consequently cultivated as the staple food of people living there [8].

Barley powdery mildew is an airborne fungal disease caused by *Blumeria graminis* f. sp. *hordei* (*Bgh*), which frequently occurs in Europe, North Africa and China [9,10,11]. It damages barley quality and results in a yield loss of generally 20% and more than 30% in outbreaks [12,13,14]. In contrast to many other plant pathogens, *Bgh* is extraordinarily adaptable, and some commonly recommended strategies involving genetic resistance may be ineffective [15]. As one of the consequences of global climate change and warming, powdery mildew has become more prevalent in the Tibet-Qinghai plateau and surrounding areas, and it urgently needs to be controlled [16,17]. Among the feasible strategies, breeding and deploying resistance varieties is the most efficient, economic and environment-friendly way to control the disease. Over 50 genes, including alleles, have been identified, and are available for resistance breeding [18]. However, the variation in the composition of the pathogen population, especially in its virulence to different genes can frustrate such effort, and the breakdown in resistance caused by such changes has caused the bottleneck of resistance breeding and deployment. To obtain virulence and genetic information of the *Bgh* populations in Tibet and the surrounding areas, 12 disease nurseries were set up in 2020 and 2021. Together with the observation of the resistance expression of barley varieties and lines bearing different resistance genes (data published elsewhere), 321 *Bgh* isolates derived from single colonies from disease samples were collected and characterized for their virulence and SNP polymorphism of genes coding for alternative oxidase (AOX), protein kinase A (PKA), and protein phosphatase type 2A (PPA) [19].

## 2. Materials and Methods

### 2.1. Lines and Varieties of Barley

Seeds of the susceptible control Hua 30 and 30 near-isogenic lines (NILs) harboring known resistance genes to *Bgh* (Table A1) were provided by Dr. Lin at the Institute of Plant Protection, China Academy of Agricultural Sciences. The susceptible variety QB01 (Zangqing 23) was a hull-less barley, provided by the Tibet Academy of Agriculture and Animal Husbandry Sciences for the isolation, purification and multiplication of the *Bgh* isolates.

### 2.2. Disease Samples and Bgh Isolates

Twelve disease nurseries including five in Tibet, three in Sichuan, two in Qinghai, and one in both Gansu and Yunnan Province (details shown in Figure A1), were set up. Seeds of each line or variety were sown in a 1.5 m long line, without replicates. The susceptible variety Hua 30 was sown to surround the disease nurseries, as the spreading line. Symptoms of powdery mildew developed at nine disease nurseries, at Qamdo (30°56′33″ N, 97°21′42″ E, 3160 m asl), Linzhi (29°43′9″ N, 94°34′46″ E, 2998 m asl), Shannan (29°10′9″ N, 91°45′55″ E, 3595 m asl), Lhasa (29°16′32″ N, 90°23′57″ E, 3662 m asl), and Rikaze (29°10′17″ N, 89°5′15″ E, 3820 m asl) in Tibet (5/5), Hezuo (35°5′53″ N, 102°54′29″ E, 2730 m asl) in Gansu (1/1), Chengdu (30°47′55″ N, 103°54′2″ E, 549 m) and Xichang (27°56′59″ N, 102°10′36″ E, 1527 m) in Sichuan (2/3) and Diqing (27°48′39″ N, 99°39′28″ E, 3280 m asl) in Yunnan (1/1). Diseased leaves were sampled from the spreading lines in the disease nurseries, and a total of 229 single-colony *Bgh* isolates were obtained and kept on the seedlings of the barley variety QB01 in test tubes of 5 cm in diameter containing sterilized soil, following the methods of Xu et al. [20].

### 2.3. Spore Preparation

To multiply the *Bgh* isolates, conidia of each isolates from the above seedlings were spread onto the seedlings of QB01 grown in pots containing 400 mL sterilized soil and then incubated in a growth chamber at 18 ± 2 °C, with continuous illumination. To prevent cross contamination, seedlings were completely encased by a glass cylinder of 10 cm in diameter, with five layers of gauze on top. When white mycelia emerged 3–5 d post inoculation (dpi), leaf segments of 5 cm in length were cut off the inoculated leaves, placed face-up onto 1% agar plates amended with 60 mg/L benzimidazole, and grown at 16 ± 2 °C, with a 16/8 h (light/dark) illumination regime. After 5–7 d of incubation, the conidia were collected onto parchment paper in the laminar flow and placed into 2.0 mL centrifuge tubes. The fresh conidia were used for the virulence assay, or first frozen in liquid nitrogen and then conserved at −80 °C for further DNA extraction [21].

### 2.4. Virulence Typing

Clumps of five germinated seeds of QB-01 and Hua 30, as well as those of 30 barley NILs, were sown in 7 rows in a 26 cm × 18 cm × 2 cm enamel tray containing sterilized soil without replicates. At the 2-leaf stage, each tray was transferred into steel cylinders of 0.4 m in diameter and 1 m in height, and inoculated with the conidia of one *Bgh* isolate by gently shaking the sporulating leaf segments. The tray was then encased in a transparent plastic bag supported by an iron framework, and incubated for 7–10 d at 18 ± 2 °C with continued illumination until susceptible QB-01 and Hua 30 showed symptoms. The infection types (ITs) of each barley NIL from different *Bgh* isolates were scored on a scale of 0 to 4, according to Jensen et al. [22], and resistance/susceptibility responses were obtained, i.e., 0–2 for resistant (R) and 3–4 for susceptible (S). The 21 selected NILs, shown in Figure 1, were clustered into groups using an octal numbering system. These groups were ranked according to the sum of three values of each group, and the resulting number (reverse-octal notation) was used to designate the pathotype of the isolates [23,24].

### 2.5. DNA Extraction and PCR Amplification

Genomic DNA of *Bgh* isolates was extracted using a Plant Genomic DNA kit DP305 (Tiangen Biotech, Beijing, China), according to the manufacturer’s instructions. Three primer sets specific for alternative oxidase (AOX; F/R-CGCATAGCCCTTTACTTAG/ATGGATTTGGGTCCTCGTT), protein kinase A (PKA; F/R-ATTTCGGTAGGGTTCATCTGG/TACCGTTCCGTCTCTTCAGG), and protein phosphatase type 2A (PPA; F/R-TAGATGGGTGGATTGAGAAC/ATCGTCAGGATCAGACCATA) were used to study the single nucleotide polymorphism (SNP) [25]. The 25 μL PCR reaction system contained 2 × Taq PCR MasterMix 12.5 μL, template DNA (10 ng/μL) 2.5 μL, with each primer (10 μmol/L) 1.0 μL, and ddH_2_O 8.0 μL. The amplification program consisted of a pre-denaturation at 95 °C for 5 min, followed by 34 cycles of denaturation at 95 °C for 25 s, annealing at 56 °C for 25 s, extension at 72 °C for 45 s, and a final extension at 72 °C for 7 min. After that, 5 μL of the PCR products were detected using 1% agarose gel electrophoresis. The PCR products with clear and bright bands and of the expected size were sent to Tsingke Biotechnology (Beijing, China) for sequencing.

### 2.6. Data Processing

The QD-SHAN in NTSYSpc 2.10s package was used for the clustering analysis of the 21 differential lines of pallas, based on the virulence assessment data of 229 *Bgh* isolates. Principal coordinate analysis (PCoA) was performed using an online tool (https://hiplot.com.cn accessed on 4 November 2022). VAT software was used to identify the *Bgh* pathotype based on the parameters related to the virulence diversity, including average relative-virulence complexity (*Rci*), gene diversity (*Hs*), genotypic diversity (*Sh*) and genetic diversity (*KWm*) [26]. DNAMAN was used to assemble the nucleotide sequences of 321 isolates, using previously published sequences as references, and the AOX-PKA-PPA sequences of the isolates was aligned and compared using the MEGA7.0. Using the Euclidean squared distance matrix in the Arlequin 3.5 package, a gene haplotype network was constructed using the PopART software, and the geographic distribution of haplotypes was processed using the ArcGIS10.2. STRUCTURE 2.3.4 under K = 1–9 was used to predict population structure, and the ΔK value was used to determine the optimal number of populations [27]. Finally, a histogram of the population-structure-related data was generated, using the Distruct software. DnaSP 5.0 [28] was used to compute diversity indices including haplotype diversity (Hd) and nucleotide diversity (Pi), to conduct neutrality tests including Tajima’s *D* as well as Fu and Li’*F** tests, and to infer population expansion. AMOVA (Analysis of Molecular Variance) in the Arlequin 3.5 package [29,30] was used to compute the genetic differentiation fixation coefficient *F*_CT_/*F*_ST_ of the inter-geographic groups/inter-populations, and to conduct a neutrality test with 1000 random permutations.

## 3. Results

### 3.1. Virulence Diversity of Bgh Populations

#### 3.1.1. Virulence Frequencies of Bgh

The virulence frequencies to different NILs as well as Hua 30 of the *Bgh* isolates from different locations were obtained as seen in Table 1. The virulence frequencies of all of the sampled *Bgh* population to P01 (*Mla1* + *MlaAl2*), P03 (*Mla6* + *Mla14*), P04A (*Mla7* + *Mlk*), P04B (*Mla7* + *MlaNo3*), P06 (*Mla7* + *M1Lg2*), P07 (*Mla9* + *Mlk*), and P08B (*Mla9*) was 0. Significant differences in virulence to the other NILs were found among different *Bgh* populations. Virulent isolates to P08A (*Mla9* + *Mlk*) and P10 (*Mla12* + *MlaEm2*) emerged only at 1 or 2 locations, while virulent isolates to P09 (*Mla10* + *MlaDu2*), P12 (*Mla22*), P15 (*MlRu2*), P18 (*Mlnn*) P20 (*Mlat* + *Mla8*), P23 (*MlLa*) and P30 were present in all of the populations. The virulence frequencies to P02 (*Mla3*), P05, P13 (*Mla23*), P16 (*Mlk*) and *mlo5* were less than 20%. The virulence frequency to the other eight resistance genes ranged from 25% to 50%, indicating that the NILs possessing these genes had moderate susceptibility (MS) or moderate resistance (MR). The genes in P09 (*Mla10* + *MlaDu2*), P12 (*Mla22*), P15 (*MlRu2*), P18 (*mlnn*), P30 and pallas were no longer efficient. Based on the resistance frequencies of different NIL lines and the clarity of their resistance genes, 21 NIls were selected as the differential lines for virulence typing (Figure 1).

#### 3.1.2. Virulence Types of Bgh

Based on the virulence spectra of 229 *Bgh* isolates to 21 barley NILs, a total of 132 pathotypes were designated (Table A2), with an average of 1.7 isolates per pathotype. Of these, 16 pathotypes had frequencies of higher than 2, and most of the isolates had one specific pathotype. Pathotypes 7.4.0.0.0.0.0, 7.3.0.0.0.0.0, 7.2.0.0.0.0.0, 7.1.0.0.0.0.0, 7.0.0.0.0.0.0, 5.4.0.0.0.0.0, 3.5.0.0.0.0.0, and 3.3.0.0.0.0.0 were common in the *Bgh* isolates from Tibet, and the pathotype 7.3.0.0.0.0.0 had the highest frequency, of 5. Pathotypes 7.3.6.5.0.0.0, 7.0.1.0.0.0.0, 6.4.2.0.1.0.0, and 6.1.2.0.0.0.0 were popular in the *Bgh* populations of Sichuan. The PCoA of the *Bgh* (Figure 2) was performed to represent the distance between the population of isolates from the nine disease nurseries, indicating that the *Bgh* populations of Diqing in Yunnan province, Hezuo in Gansu province and Chengdu and Xichang in Sichuan province had similar virulence spectra, which were different from those of the four Tibetan *Bgh* populations.

#### 3.1.3. Virulence Diversity of Bgh

The virulence diversity of the *Bgh* populations from nine disease nurseries in Tibet, Sichuan, Yunnan, and Gansu were evaluated. As shown in Table 2, the *Bgh* population at Diqing in Yunnan had the highest virulence complexity, with an *Rci* value of 0.363, whereas that of Rikaze in Tibet was the lowest (0.160). The *Bgh* populations at Diqing in Yunnan and Xichang in Sichuan had genetic diversity value of *KWm* at 0.385, while the populations at Shannan in Tibet had the lowest *KWm* (0.143). The relationship between the populations was analyzed based on the genetic distance Nei’s (*N*) and mean character difference (*MCD*) (Table 3). The genetic distance between the populations at Diqing and Shannan was greatest for *N* (0.192), and the *MCD* value (0.292) between these two population was also the highest. The *N* and *MCD* values between the populations at Lhasa, Shannan, and Qamdo in Tibet were the lowest, and were 0.017 and 0.082, respectively.

### 3.2. Molecular Genetic Structure of the Bgh Populations

#### 3.2.1. Haplotype Composition of Different Bgh Population

After the sequences of the genes of AOX, PKA and PPA in the genome of 321 isolates were organized through alignment analysis and assembled, the length of the sequence containing polymorphic sites was 2196 bp for all the isolates. Excluding gaps and three deletion sites, a total of seven polymorphic sites, two singleton variable sites (at positions 394 and 1883), and five parsimony informative sites (at positions 238, 738, 1279, 1539, and 1569) were found in the sequences. In total, nine haplotypes (H1–H9) were detected (Table 3). The haplotype network and the geographical distribution of the isolates (Figure 3 and Figure 4) indicated that, among the nine haplotypes, two haplotypes (H1 and H4) were dominant and accounted for 29.6% and 59.2% of the isolates, respectively. H1 existed in populations in Sichuan, Gansu, and Yunnan and the population at Qamdo in Tibet, while H4 haplotype existed in all of the five Tibetan populations and in the population at Chengdu in Sichuan.

The genetic structure of the *Bgh* populations was analyzed employing STRUCTURE software, and the K values were set from 1 to 9, to predict the optimal population structure. When K = 2, the ΔΚ value was the largest, suggesting that the optimal number of sub-populations was 2 for the grouping (Figure 5). The nine *Bgh* populations were classified into two sub-populations, of which subpopulation 1 involved all of the Tibetan populations, and subpopulation 2 involved the populations from the surrounding Sichuan, Gansu, and Yunnan. Populations from closer geographic locations had higher genetic-structure similarity.

#### 3.2.2. Genotype Diversity of Bgh Populations

The total haplotype diversity (represented by Hd) of the investigated *Bgh* populations was 0.564, and the average nucleotide diversity (represented by π) was 0.00034. The Hd value ranged from 0 to 0.6485, and the π value ranged from 0 to 0.00049, illustrating the large variation between the populations. The population at Qamdo in Tibet had the highest Hd (0.64848), followed by the population at Hezuo in Gansu and Xichang in Sichuan, which were 0.34762 and 0.31169, respectively. The populations at Lhasa, Linzhi, and Shannan in Tibet and Diqing in Yunnan showed the lowest haplotype and nucleotide diversity, and only one haplotype was detected in each of these populations There were two haplotypes in the population at Rikaze (Table 4 and Table 5).

The AMOVE analysis showed (Table 6) that inter-group genetic variation accounted for 54.68%, inter-population genetic variation accounted for 23.90% and intra-population genetic variation accounted for 21.42%. The inter-group genetic differentiation coefficient, *F*_CT_, was 0.54681, and the average inter-population genetic differentiation coefficient, *F*_ST_, was 0.69714, indicating that larger genetic differentiation of *Bgh* populations mainly existed between groups. There was a higher level of genetic differentiation and a lower level of gene exchange between the populations. The neutrality test showed that the Fu’s Fs value was −2.992, and the Tajima’s D value was −0.65769. These were non-significant (*p* > 0.10), indicating that the *Bgh* populations followed the neutral model of evolution, and there were many loci with low-frequency alleles. The Fu and Li’ D (−0.89307) and Fu and Li’*F* (−0.97079) were also non-significant (*p* > 0.10), indicating that the *Bgh* populations were affected by selection and recombination.

## 4. Discussion

The differentiation of *Bgh* into different populations of distinct virulence and genetic composition has been reported globally [3,11,12,13,14,15]. In China, it has been found that there has been a recent change in the virulence frequency of *Bgh* to resistance genes and or gene combinations, which shows an overall trend toward higher pathotype diversity and virulence complexity [12,21,31]. The *Bgh* populations in central Europe also appear to exhibit a similar trend [32,33,34]. The isolates from Yunnan and Zhejiang had similar virulence profiles, but differed from those identified in Tibet [21]. In this research, disease samples were mainly collected from the cultivation areas of hull-less barley, except for Xichang and Chengdu in Sichuan. There were still 11 NILs with a virulence frequency of less than 10.0%. Eight out of the seventeen NILs expressing high resistance in the research of Wang et al. [21] were no longer highly resistant. This may be due to the differences in the sampling sites and possible breakdown of resistance in Tibet. From the result of the virulence assay of the *Bgh* population and disease incidence at different disease nurseries (unpublished data), resistance genotypes *Mla6* + *Mla14*, *Mla7* + *Mlk*, *Mla7* + *Mla* (*No3*), *Mla7* + *M1* (*Lg2*), *Mla9*, *Mla9* + *Mlk* and *Mla12* + *MlaEm2* are highly resistant to *Bgh* in Tibet and the surrounding areas. Inside Tibet, *Mlra*, *Mla23*, *Mlk1*, *Mlg* + *Ml(CP)* and the broad-spectrum resistance gene, *mlo5*, are also still resistant to powdery mildew. In order to obtain higher and more durable resistance, it would be valuable to pyramid such genes in combination with *Mlo*, together with the continuous monitoring of the changes in the virulence frequencies of different *Bgh* populations.

While barley is newly cultivated in Xichang and rare in Chengdu and Xining, the hull-less barley has long been introduced, first into the southern and eastern areas of the Qinghai-Tibet plateau, and has experienced a process of adaptation to the environment of varying altitudes [35]. Although it is in the recent two decades that powdery mildew has been reported to cause severe loss in Tibet [21], the incidence of the disease was documented as early as in 1987 [36]. Powdery mildew disease has not been observed in the disease nurseries set up at Daofu in Sichuan, Haibei and Xining in Qinghai. Severe powdery mildew was observed in the disease nursery at Hezuo in Gansu in 2019, but only a small number of colonies were observed in late September 2020, during 2020–2022. Obviously, the conclusion after the genetic analysis in this research, that the *Bgh* population had experienced a recent expansion, is congruent with the recent disease epidemic in the Tibet-Qinghai Plateau.

Both geographic isolation and long-distance migration had impacts on the population diversity of *Blumeria graminis*, and [21] has revealed the difference in *Bgh* genotypes between Tibet, Yunnnan, Sichuan in the southwest and Zhejiang in the southeast of China. The *Bgh* genotypes in this research revealed two centers of genetic diversity, i.e., Qamdo and Hezuo. The prevalence of H1 haplotypes at Qamdo and the locations surrounding Tibet suggested the possibility of the pathogen isolates being introduced into Tibet from other areas. When populations inside Tibet were looked at closely, *Bgh* seemed to have originated in the areas of Qamdo and expanded to the western areas of higher altitudes and lower temperature and humidity at Linzhi, Lhasa, Shannan and Rikaze. The introduction of barley into the Tibet-Qinghai plateau has resulted in the loss of most of the genetic diversity of the crop [35]. It is reasonable to postulate that the loss of resistance genes also happened during such a process due to the lack of disease pressure on the high and arid plateau. The results of virulence typing both in this research and in that of Wang et al. [21] revealed the weaker virulence of the Tibet *Bgh* population compared with that in the other areas of China. In contrast to the less diversified genetic structure than that of *Bgh* at Qamdo, the virulence become stronger in the pathogen population at the western sites in Tibet. This could be explained by the more intensive cultivation of barley with newly introduced resistance genes in commercial cultivars in the latter locations.

## Figures and Tables

**Figure 1 jof-09-00363-f001:**
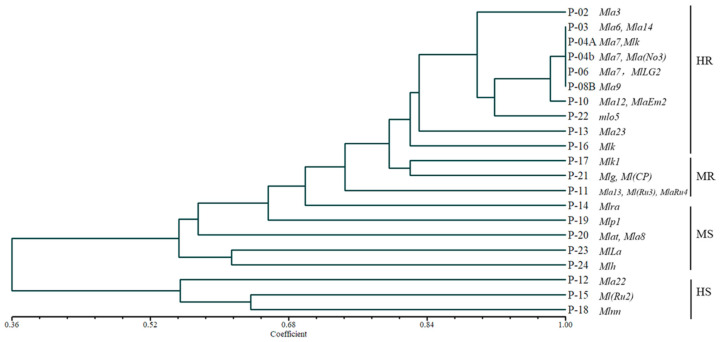
Dendrogram based on the cluster analysis of infection types of 21NILs of barley variety pallas from 229 *Bgh* isolates, using the QD-SAHN program in NTSYSpc 2.10s.

**Figure 2 jof-09-00363-f002:**
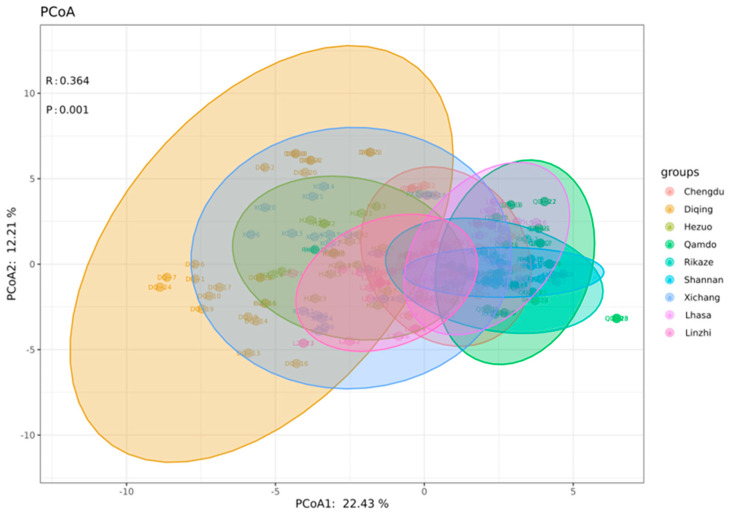
PCoA of the virulence variability of *Bgh* isolates from nine disease nurseries in Tibet, Sichuan, Yunnan, and Gansu.

**Figure 3 jof-09-00363-f003:**
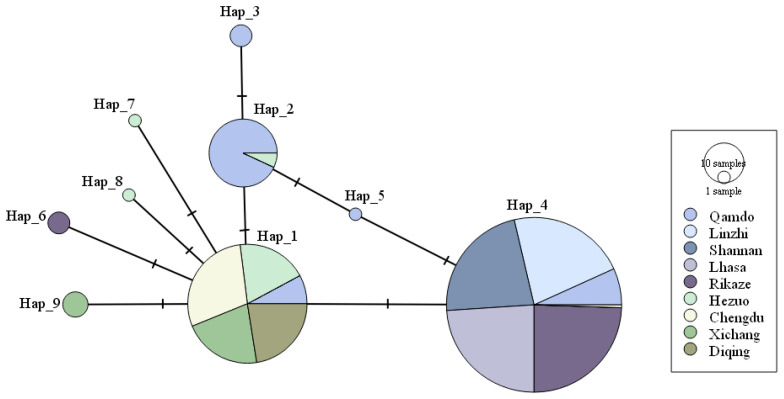
PopART Network of nine haplotypes of *Blumeria. graminis* f. sp. hordei.

**Figure 4 jof-09-00363-f004:**
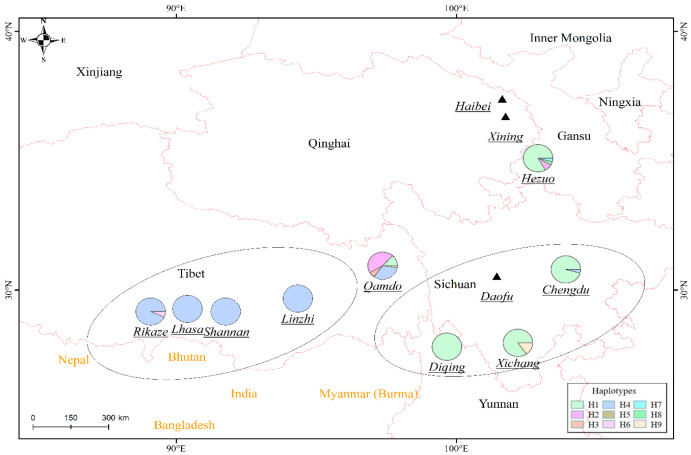
Distribution of the test isolates for haplotypes of *Blumeria. graminis* f. sp. hordei. (Note: The base drawing is from the website of the Ministry of Natural Resources (http://www.mnr.gov.cn accessed on 9 November 2022), the review drawing number is GS (2020) 4621, and the base drawing has not been modified).

**Figure 5 jof-09-00363-f005:**
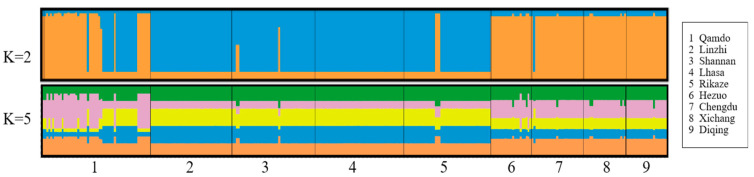
Genetic structure map inferred by STRUCTURE analysis consisting of *Blumeria. graminis* f. sp. *hordei* from nine populations.

**Table 1 jof-09-00363-t001:** Virulence frequencies of *Blumeria. graminis* f. sp. *hordei* from nine populations isolates on 30 barley NILs.

Variety	Resistance Gene(s)	Virulence Frequency (%)									
		QD ^a^	LZ	SN	LS	RKZ	HZ	CD	XC	DQ	Total
P01	*Mla1, Mla(Al2)*	0	0	0	0	0	0	0	0	0	0
P02	*Mla3*	0	35.71	0	0	0	16.67	11.54	25.00	0	9.88
P03	*Mla6, Mla14*	0	0	0	0	0	0	0	0	0	0
P04A	*Mla7, Mlk*	0	0	0	0	0	0	0	0	0	0
P04B	*Mla7, Mla(No3)*	0	0	0	0	0	0	0	0	0	0
P05	*/*	0	21.43	0	0	0	16.67	7.69	41.67	58.33	16.20
P06	*Mla7, Ml(Lg2)*	0	0	0	0	0	0	0	0	0	0
P07	*Mla9, Mlk*	0	0	0	0	0	0	0	0	0	0
P08A	*Mla9, Mlk*	0	0	0	4.55	0	0	0	0	0	0.50
P08B	*Mla9*	0	0	0	0	0	0	0	0	0	0
P09	*Mla10, MlaDu2*	60.00	100	66.67	100	70.00	91.67	76.92	100	100	85.03
P10	*Mla12, MlaEm2*	6.67	0	0	0	0	0	0	0	0	0.74
P11	*Mla13, Ru3, Ru4*	40.00	0	0	54.54	3.70	0	23.08	54.00	33.33	23.18
P12	*Mla22*	53.33	100	83.33	63.64	77.78	75.00	65.38	33.00	58.33	67.75
P-13	*Mla23*	0	21.43	16.67	13.64	0	12.50	19.23	4.17	54.17	15.76
P-14	*Mlra*	0	21.43	0	0	0	54.17	65.38	46.00	66.67	28.18
P15	*Ml(Ru2)*	60.00	78.57	83.33	81.82	44.44	83.33	76.92	92.00	91.67	76.90
P16	*Mlk*	0	21.43	0	0	7.41	0	15.38	33.00	6.25	9.27
P17	*Mlk1*	0	7.14	0	0	3.70	62.50	0	37.50	75.00	20.65
P18	*Mlnn*	60.00	64.29	71.00	72.73	48.15	83.33	84.62	50.00	41.67	63.98
P19	*Mlp1*	0	75.00	0	0	7.41	8.75	7.69	71.00	91.67	29.06
P20	*Mlat, Mla8*	73.33	21.43	66.67	72.73	14.81	25.00	23.08	16.67	16.67	36.71
P21	*Mlg, Ml(CP)*	0	0	0	18.18	0	54.17	23.08	33.00	58.33	20.75
P22	*mlo5*	0	0	0	4.55	22.22	16.67	0	21.00	12.50	8.55
P23	*MlLa*	46.67	71.43	91.67	50.00	40.74	29.17	15.38	54.00	33.33	48.04
P24	*Mlh*	33.33	27.43	66.67	36.36	59.26	12.5	30.77	0	58.33	36.07
P29	*/*	0	0	0	0	0	0	0	0	0	0
P30	*/*	73.33	92.86	91.67	81.82	85.19	100	92.31	100	100	90.80
P31	*/*	0	0	0	0	0	0	0	0	0	0
Pallas	*Mla8*	100	100	100	100	100	100	100	100	100	100

^a^ QD, Qamdo; LZ, Linzhi; SN, Shannan; LS, Lasha; RKZ, Rikaze; CD, Chengdu; XC, Xichang; DQ, Diqing.

**Table 2 jof-09-00363-t002:** Virulence diversity parameters of nine *Bgh* populations.

	QD	LZ	SN	LS	RKZ	HZ	CD	XC	DQ
Number of isolates	30	28	24	22	27	24	26	24	24
Number of pathotypes	13	18	10	18	15	16	21	22	19
Average virulence complexity per isolate (*Ci*)	3.733	5.393	5.000	4.682	3.370	6.042	4.808	5.625	7.625
Relative virulence complexity per isolate (*Rci*)	0.178	0.257	0.238	0.223	0.160	0.288	0.229	0.268	0.363
Genotypic diversity (sh)	0.742	0.846	0.670	0.903	0.797	0.836	0.912	0.964	0.891
Gene diversity (*HS*)	0.162	0.165	0.107	0.173	0.165	0.209	0.205	0.261	0.256
Genetic diversity (*KWm*)	0.248	0.248	0.143	0.264	0.250	0.298	0.267	0.385	0.385

**Table 3 jof-09-00363-t003:** Nei’s (*N*) standard genetic distances and mean character difference (*MCD*) between nine *Blumeria graminis* f. sp. *hordei* populations.

Populations	QD	LZ	SN	LS	RKZ	HZ	CD	XC	DQ
QD	-	0.184 ^2^	0.103	0.184	0.111	0.237	0.144	0.18	0.282
LZ	0.105 ^1^	-	0.147	0.182	0.157	0.151	0.162	0.144	0.234
SN	0.044	0.08	-	0.082	0.119	0.220	0.170	0.240	0.292
LS	0.017	0.102	0.041	-	0.139	0.151	0.162	0.169	0.234
RKZ	0.048	0.074	0.057	0.062	-	0.201	0.156	0.184	0.237
HZ	0.14	0.07	0.143	0.119	0.114	-	0.157	0.127	0.151
CD	0.072	0.085	0.098	0.06	0.071	0.083	-	0.152	0.206
XC	0.101	0.061	0.134	0.086	0.093	0.052	0.061	-	0.155
DQ	0.185	0.119	0.192	0.158	0.151	0.07	0.112	0.057	-

^1^ Nei’s (N) values: below diagonal; ^2^ Mean character difference (MCD) values: above diagonal.

**Table 4 jof-09-00363-t004:** Distribution of *Blumeria. graminis* f. sp. *hordei* single-nucleotide polymorphism haplotypes.

Haplotype	AOX ^a^		PKA		PPA			QD	LZ	SN	LS	RKZ	HZ	CD	XC	DQ
H1	A ^b^	G	A	G	A	C	C	6					20	23	22	24
H2			G					21					2			
H3			G		G			3								
H4						A		15	43	45	44	39		1		
H5			G			A		1								
H6	T											3				
H7							G						1			
H8		A											1			
H9				A											4	
Total								46	43	45	44	42	24	24	26	24

^a^ AOX = alternative oxidase, PKA = protein kinase A, and PPA = protein phosphatase type 2A. ^b^ Most frequent or consensus nucleotide for each variable position.

**Table 5 jof-09-00363-t005:** Genetic diversity of multi-locus sequence haplotypes of *Blumeria. graminis* f. sp. *hordei* isolates from nine populations.

Populations	Number of Isolates	Haplotypes Diversityjof (Hd)	Nucleotide Diversity (π)	Average Number of Difference (K)
QD	46	0.6485	0.00049	1.00735
LZ	43	0.000	0.00000	0.00000
SN	45	0.000	0.00000	0.00000
LS	44	0.000	0.00000	0.00000
RKZ	45	0.1151	0.00001	0.23020
HZ	24	0.3477	0.00017	0.37143
CD	24	0.3117	0.00007	0.07407
XC	26	0.1453	0.00014	0.31169
DQ	24	0.0000	0.00000	0.00000
Total	321	0.5780	0.00034	0.74034

**Table 6 jof-09-00363-t006:** Analysis of molecular variance (AMOVA) for *Blumeria. graminis* f. sp. *hordei* isolates from nine populations.

Source of Variance	d.f	Sum of Squares	Variance Components	Percentage of Variation (%)	Fixation Indices
Among groups	1	45.268	0.31240 Va	54.68	*F*_CT_ = 0.54681
Among populations	8	35.626	0.13655 Vb	23.90	*F*_ST_ = 0.69714
Within populations	313	38.302	0.12237 Vc	21.42	
Total	322	119.195	0.57132		

## Data Availability

All data supporting the findings of this study are available within the paper.

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
