# Peer review of "Virulence and Genetic Types of Blumeria graminis f. sp. hordei in Tibet and Surrounding Areas"

_jof, 2023, doi:10.3390/jof9030363_

Round 1

Reviewer 1 Report

Overall the manuscript represents an interesting study in barley powdery mildew from Tibet and the surrounding areas. However, it would benefit from careful proof-reading e.g. 'Amova' and 'Amove'. General grammar and spelling checks should also be conducted throughout.

In the Introduction, a range of references are given for reported yield losses. These should be more precise. 

In the Materials and Methods, what exactly is a 'hill'?. Do the authors mean "hill plots', which are usually groups of plants grown outside? it's also not clear how may replicates of each differential line were used.   

Author Response

Dear Reviewer:

Thank you for your kind words and apologize for our unclear description. We have made the followed changes.

Comments 1 :  Overall the manuscript represents an interesting study in barley powdery mildew from Tibet and the surrounding areas. However, it would benefit from careful proof-reading e.g. 'Amova' and 'Amove'. General grammar and spelling checks should also be conducted throughout.

Response 1 :The manuscript has been re-written and attention has been paid to the grammar and spelling checks.

Comments 2 :  In the Introduction, arange of references are given for reported yield losses. These should be more precise. 

Response 2:In the introduction, the d the changes in the distribution of barley in China has been specially described and the long history and distinction of Tibet barley to other areas in China has been emphasized.

Comments 3 :  In the Materials and Methods, what exactly is a 'hill'?. Do the authors mean "hill plots', which are usually groups of plants grown outside? it's also not clear how may replicates of each differential line were used.

Response 3 :  “Hill” has been changed to clump. The lines and varieties used in the study has been described in the part of Materials and Methods.

Your Sincerely,

Wang Yunjing

Reviewer 2 Report

The work by Wang et al. (Virulence and Genetic typing reveal the recent expansion of Blumeria graminis f. Sp. Hordei in Tibet China),

1.     Title should be rewritten.

2.     Abstract is too long. It should be shortened and rewritten.

3.     Introduction is poor. It should be extended and rewritten. The objective of the study must be added to the last paragraph of the introduction.

4.     The aim of the study is missing.

5.     The manuscript is not well-organized.

6.     The manuscript requires detailed editing in terms of language.

It is almost impossible to follow and understand the general concept of the study. Most parts of the reports must be reconsidered and rewritten. Therefore, I recommend the rejection of the report in its present form.

Author Response

Dear Reviewer,

Thank you very much for your professional review, which was a great help in revising the article. We have made the following changes according to your comments.

Comments 1:  Title should be rewritten.

Response 1:  Tittle has been changed as “Virulence and Genetic Types of Blumeria graminis f. sp. hordei in Tibet and Surrounding Areas”

Comments 2:  Abstract is too long. It should be shortened and rewritten.

Response 2:  Thank you very much for your professional review, The abstract has been shortened and re-written.

Comments 3:  Introduction is poor. It should be extended and rewritten. The objective of the study must be added to the last paragraph of the introduction.

Response 3:  The introduction has been re-written and the objective of the study has been added to the last paragraph of the introduction.

Comments 4:  The aim of the study is missing.

Response 4:  It has been added in the introduction.

Comments 5:  The manuscript is not well-organized.

Response 5:  Your comments are very important to the revision of the article, and we look forward to your comments. The whole manuscript has been re-written and organized.

Comments 6:  The manuscript requires detailed editing in terms of language.

Response 6:  Special checks of spelling and grammar have been made. Thank you again.

Your Sincerely,

Wang Yunjing

Reviewer 3 Report

Comments:

Powdery mildew is serious threat in barley production worldwide. In this manuscript, Wang et al. reported that the difference in genotypes of between Tibetan Bgh population and the neighboring populations in Yunnan and Sichuan as well as in southeastern areas of China. The authors have collected Bgh samples from Qinghai different Tibet Plateau areas, and analyzed the virulence diversity of Bgh populations, Molecular genetic structure of the Bgh populations. The result of virulence and genetic typing proves that the Bgh population in Tibet has experienced a recent expansion accompanied by a loss of genetic diversity and an increase in virulence. In general, this is a report with considerable workload. The author has carried out a lot of research and analysis on the local agricultural demand. The data is reliable, the analysis is rigorous, the full text structure is reasonable, and the discussion is persuasive. However, there are still some suggestions for the authors that should be addressed.

Major points:

1. What’s the difference between Tibet area and other area in China?Is Tibet an independent disease epidemic area?

2. How does Bgh population change comparing the past and the present Tibet area? I suggest the author compare your conclusion with other published articles in the discussion section.

3. The Qinghai Tibet Plateau is a place where the altitude changes dramatically. You'd better add its longitude, latitude and altitude when describing each place.

4. Have you investigation any Tibet local commercial varieties cultivar resistance against these isolates? Is there new race threats Tibet barley production?

Minor points:

In Abstract section:

Line 8: It was well known that Qingke (Tibetan hulless barley) has long been cultivated on the Tibetan Plateau. The authors need to clarify the plant used in the research

In Introduction section:

Line 35-52: The Introduction is too brief and the research background could not be well introduced. It is difficult to publish directly at present.

In Materials and Methods section:

Line 58: Please offer geographic coordinates of sample sites

Line 69: “400ml” should be “400 mL”. Please check similar problems

Line 73, “5cm” should be “5 cm”, There should be spaces between numbers and units. Same mistakes also in L75, L87, L89...

In Results section:

Line 158, “Bgh” should “Bgh”.

Line 171-174: The authors need a detailed description for PCoA results.

Line 173: check “Sichuan province

In Discussion section:

Line 277: “Wang er al” should be “Wang et al”

Line 300: check “and or”

In References section:

The references did not conform to the specifications.

https://www.mdpi.com/journal/jof/instructions#preparation

Author Response

Dear Reviewer,

Thank you very much for your time involved in reviewing the manuscript and your very encouraging comments on the merits. We appreciate your clear and detailed feedback and hope that the explanation has fully addressed all of your concerns.

Comments:

Major points:

Comments 1: What’s the difference between Tibet area and other area in China?Is Tibet an independent disease epidemic area?

Response 1:  The results of virulence and genetic types reveal difference of Tibetan population of Blumeria graminis f. sp. hordei from that in surrounding areas, and Qamdo in Tibet was considered to be the center of connection between Tibet and its surrounding areas.

Comments 2:  How does Bgh population change comparing the past and the present Tibet area? I suggest the author compare your conclusion with other published articles in the discussion section.

Response 2:  It is only recently that mildew has been eye-spotted and causes losses in Tibet while the disease has been a severe problem in other areas China. The comparison has been made and discussed.

Comments 3:  The Qinghai Tibet Plateau is a place where the altitude changes dramatically. You'd better add its longitude, latitude and altitude when describing each place.

Response 3: It has been added in the materials and methods.

Comments 4:  Have you investigation any Tibet local commercial varieties cultivar resistance against these isolates? Is there new race threats Tibet barley production?

Response 4:  The resistance or susceptibility of major commercial varieties form Tibet, Sichuan and Yunnan has also been observed in the disease nurseries. But they were not inoculated by Bgh isolates. The results of this study was compared to that of Wang et al. (2019) and changes in the virulence could be observed. Due to short history of the disease epidemic in Tibet, the emergence of new races has been further monitored in the futural researches.

Minor points:

In Abstract section:

Line 8:  It was well known that Qingke (Tibetan hulless barley) has long been cultivated on the Tibetan Plateau. The authors need to clarify the plant used in the research.

Responses:  The lines and varieties used in the study has been described in the part of Materials and Methods.

In Introduction section:

Line 35-52:  The Introduction is too brief and the research background could not be well introduced. It is difficult to publish directly at present.

Responses:  Thank you very much for your professional review, we have tried our best to revise our manuscript according to the comments, while adding the introduction section.

In Materials and Methods section:

Line 58:  Please offer geographic coordinates of sample sites

Line 69: “400ml” should be “400 mL”. Please check similar problems

Line 73, “5cm” should be “5 cm”, There should be spaces between numbers and units. Same mistakes also in L75, L87, L89...

Responses:  The sampl sites geographic coordinates has been added in the materials and methods.

In Results section:

Line 158, “Bgh” should “Bgh”.

Line 171-174: The authors need a detailed description for PCoA results.

Line 173: check “Sichuan province

Responses:  We thoroughly reviewed the manuscript and made appropriate changes.

In Discussion section:

Line 277: “Wang er al” should be “Wang et al”

Line 300: check “and or”

Responses:  We are very sorry for the problem caused by our negligence. Special checks of spelling has been made.

In References section:

The references did not conform to the specifications.

Responses:  The references have been revised. Thank you again.

Your Sincerely,

Wang Yunjing

Round 2

Reviewer 2 Report

After the revision, the authors significantly improved the quality of the manuscript. I recommend the publication of the report.